

# Enhancing restoration success of rare plants in an arid-tropical climate through water-saving technologies: a case study of *Scalesia affinis* ssp. *brachyloba* in the Galapagos Islands

Esme Plunkett[1], Luka Negoita[1], Christian Sevilla[2], Nicolás Velasco[1,3] and Patricia Jaramillo Díaz[1,4]

[1] Charles Darwin Foundation, Puerto Ayora, Galapagos Islands, Ecuador
[2] Galapagos National Park Directorate, Puerto Ayora, Galapagos Islands, Ecuador
[3] Institute for Evolutionary Life Sciences, University of Groningen, Groningen, Netherlands
[4] Department of Botany and Plant Physiology, Universidad de Málaga, Málaga, Spain

Corresponding author
Patricia Jaramillo Díaz,
patricia.jaramillo@fcdarwin.org.ec

## ABSTRACT

Arid tropical archipelagos, such as the Galapagos Islands, host a high concentration of endemic plant species, many of which require restoration intervention to recover from past environmental degradation. Water-saving technologies (WSTs) have potential for hastening restoration by providing plants with additional water during the early stages of growth. However, it remains unclear whether such technologies provide an advantage for plant species of arid-tropical regions. This study examined the effect of the water-saving technology Groasis Waterboxx® (Groasis) on the rare endemic plant species *Scalesia affinis* ssp. *brachyloba* during early stages of restoration. Survival was monitored for 374 individuals planted across six sites on Santa Cruz Island, Galapagos (326 with technology and 48 as controls). Kaplan-Meier survival analysis showed that the use of Groasis reduced mortality during the first two years of the seedling survival. A mixed-effect logistic regression that modelled plant survival as a function of total precipitation, maximum temperature, and WST treatment (Groasis and no-technology control) found that despite low overall survival rates, plants grown with Groasis exhibited a three-fold higher predicted survival by the end of the 3.7 year duration of the study. Finally, through a resampling method, we demonstrate that the effect of the WST treatment is not dependent on the unbalanced design typical of a restoration project framework. We conclude that water-saving technologies such as the Groasis Waterboxx® can enhance survival of rare plant species such as *S. affinis* ssp. *brachyloba* in restoration programs in arid-tropical regions.

## INTRODUCTION

Despite occupying less than 5% of the global terrestrial area, oceanic islands are home to >20% of the planet's terrestrial plant and vertebrate species (*Courchamp et al., 2014*). The

Galapagos archipelago and its flora are no exception, with 37% of its plant taxa as endemic (*Tye & Francisco-Ortega, 2011*). Despite its globally recognized significance, the Galapagos flora is currently under high levels of degradation (*Jaramillo et al., 2013*) and extinction risk (*Tye, 2007*) which has led to Galapagos flora becoming the focus of major restoration activities in the last decade (*Jaramillo et al., 2020*). In the Galapagos, as in other arid tropic regions, dry forest species are some of the most threatened species due to land-use change and stochastic rainfall seasonality, which can undermine regeneration (*Myers, Mittermeier & Mittermeier, 2000*; *Khurana & Singh, 2001*; *Tye & Francisco-Ortega, 2011*; *Gillespie et al., 2013*).

The scarcity of water and high summer temperatures in arid environments are some of the main challenges of restoration programs in these climates (*Cabin et al., 2002*; *Will et al., 2013*). Although tropical climates have wet summers, climate change has altered rainfall predictability, making plants more prone to senesce when water is scarce (*Nieuwolt, 1991*; *Anderegg, Anderegg & Berry, 2013*). Water-saving technologies (WSTs) (*i.e.,* equipment or tools that can reduce the changing levels of water balance in soil) can improve seedling survival by providing regular supplies of water and have been useful in restoration programs in other arid regions around the world (*Jaramillo et al., 2020*). WSTs have the potential to accelerate Galapagos plant species' restoration by making them less susceptible to rainfall seasonality. For example, given that water limitation has been identified as a key driver of seedling survival, preliminary restoration activities with these technologies have shown promising results (*Tapia et al., 2019*; *Negoita, Gibbs & Jaramillo, 2021*).

We focused on *Scalesia affinis* ssp. *brachyloba* as a model taxon to investigate the effect of WSTs on improving species restoration. In 2013, the Charles Darwin Foundation (CDF), *via* the Galapagos Verde 2050 (GV2050) program, began an initiative to increase populations of *Scalesia* spp., with a particular focus on *S. affinis* ssp. *brachyloba* which only grows in arid-tropical conditions (*Beck et al., 2018*) in the south-west border of Santa Cruz Island (*Harling, 1962*). The species' has evergreen leaves and a long taproot, functional characteristics for a dry environment (*Atkinson, Jaramillo & Tapia, 2009*), making it an ideal candidate for testing the potential value of WSTs. Results from this study may also be useful for understanding other species in this genus. *Scalesia* (Asteraceae) is the most diverse of seven endemic genera of the Galapagos, with 15 species and more than 20 taxa, and is often cited as a prominent example of adaptive radiation (*Adsersen & Svendsen, 1986*; *Eliasson, 1974*; *Tye, 2003*) from colonizing barren lava fields, to forming a forest canopy (*Hamann, 1979a*; *Itow, 1995*; *Kelager & Philipp, 2008*; *Atkinson, Jaramillo & Tapia, 2009*; *Watson et al., 2009*). Many *Scalesia* spp. are threatened by anthropogenic habitat changes, introduced species, and the changing climate (*Lawesson, 1986*; *Caujapé-Castells et al., 2010*). The population of *S. affinis* ssp. *brachyloba* (hereafter *S. brachyloba*) on Santa Cruz Island has dramatically declined (*Atkinson, Jaramillo & Tapia, 2009*). The historical distribution of the taxa was in the same area where the human settlements have established and expanded on Santa Cruz Island (*Harling, 1962*; *Watson et al., 2009*; *INEC, 2015*). By the year 2000 the population of *S. brachyloba* within the main town of Puerto Ayora was recorded at five individuals (*Nielsen, 2004*; *Kelager & Philipp, 2008*), and by 2007 all areas with previously large populations of *S. brachyloba* had seen population declines (*Jaramillo,*

*2007*). Low natural and assisted regeneration of the species hinders its restoration. For example, the species may reach only 17% survival 3 months after germination, and 19% one year after transplanting to the wild (*Atkinson, Jaramillo & Tapia, 2009*).

Here we aim to use *S. brachyloba* both as a study taxon for evaluating the effect of a water-saving technology (Groasis Waterboxx®, hereafter just "Groasis") on dry forest species, but results may also inform future restoration efforts of this species on Santa Cruz Island, Galapagos. Through this process we also account for the influence of the two main factors that shape arid-tropical climates: precipitation and maximum temperature.

## MATERIALS & METHODS

Through regular monitoring over six years, we collected data on the survival of *S. brachyloba* seedlings that were planted (under National Park permission N° PC-10-21) using treatments that included Groasis and controls (no technology). We also used available data from associated environmental conditions (*Jaramillo et al., 2020*).

### Study area, sites, and target species

Our study sites were based on the southwest border of Santa Cruz Island (−0.7351, −90.3089)—the second largest island on the Galapagos archipelago (*Helsen et al., 2009*), in the dry lowland zone of the island. The climate in this zone is arid tropical with a median annual rainfall of 227 mm, though 71% of this precipitation occurs during the wet season months of January through May and mean monthly temperature ranging from 19° to 30.5 °C (*Beck et al., 2018*; *Trueman & D'Ozouville, 2010*). Vegetation community is mostly xeric, composed by *S. affinis*, *Oputia echios* and *Croton scouleri* (*Huttel, 1986*). Soils are mainly clay or clay-loam (*Hengl et al., 2017*). We studied *S. brachyloba* at six sites which totaled an area of one hectare and were all locations where *S. brachyloba* had been historically found—locations that were confirmed by samples in the CDF Herbarium (*Jaramillo et al., 2018*), observations, and photographs (*Jaramillo, 2007*; *Atkinson, Jaramillo & Tapia, 2009*) (Table 1, Fig. 1). Two sites correspond to natural populations: one site was an enclosed area of national park land in an area called 'Garrapatero' in the southeast of the island and another site called 'Mirador' was on public land in Puerto Ayora (*Jaramillo, Tapia & Tye, 2018*). The other four sites were all small gardens located in Puerto Ayora, Galapagos at the Biosecurity Agency (GBA), the Galapagos National High School, the offices of the Galapagos National Park Directorate (GNPD), and the Charles Darwin Research Station (CDRS) (Fig. 1).

### Plantings and water-saving technology

*S. brachyloba* were germinated in the CDRS laboratories from seeds collected from naturally occurring individuals, then transferred to pots kept in a shade-house and monitored for 12 weeks, before being planted across the study sites. From 2013 to 2017, 374 individuals were planted at the six study sites: Garrapatero ($n = 180$), Mirador ($n = 128$), GBA garden ($n = 31$), CDRS garden ($n = 25$), Galapagos National School ($n = 4$), and GNPD garden ($n = 6$). Since the study was conducted within a restoration framework, plants that died were replaced with new seedlings during subsequent monitoring trips. Individuals were planted
**Table 1  Planting age summary of *Scalesia affinis* ssp. *brachyloba* by study site, treatment and planting date.**

| Site | Treatment | Count | Mean | | SD | Median | CI_Lower | CI_Upper |
|------|-----------|-------|------|---|----|--------|----------|----------|
| GBA | Control | 1 | 124,0 | ± | NA | 124 | NA | NA |
| | Groasis | 30 | 510,7 | ± | 560,6 | 124 | 301,4 | 720,1 |
| CDF | Control | 3 | 456,7 | ± | 456,8 | 305 | 0,0 | 1591,4 |
| | Groasis | 22 | 650,2 | ± | 539,6 | 407 | 410,9 | 889,4 |
| Galapagos College | Groasis | 4 | 1277,0 | ± | 8,0 | 1273 | 1264,3 | 1289,7 |
| Garrapatero | Control | 25 | 60,6 | ± | 29,3 | 42 | 48,6 | 72,7 |
| | Groasis | 155 | 350,8 | ± | 255,0 | 297 | 310,4 | 391,3 |
| GNP | Control | 1 | 1352,0 | ± | NA | 1352 | NA | NA |
| | Groasis | 5 | 844,2 | ± | 564,2 | 1130 | 143,7 | 1544,7 |
| Mirador | Control | 18 | 131,9 | ± | 308,5 | 46 | 0,0 | 285,3 |
| | Groasis | 110 | 625,3 | ± | 476,4 | 583 | 535,3 | 715,3 |

**Notes.**

Count = number of plants used. Summaries are mean planting age (days ± SD) and median planting age plus the lower and upper limit for its 95% confidence interval.

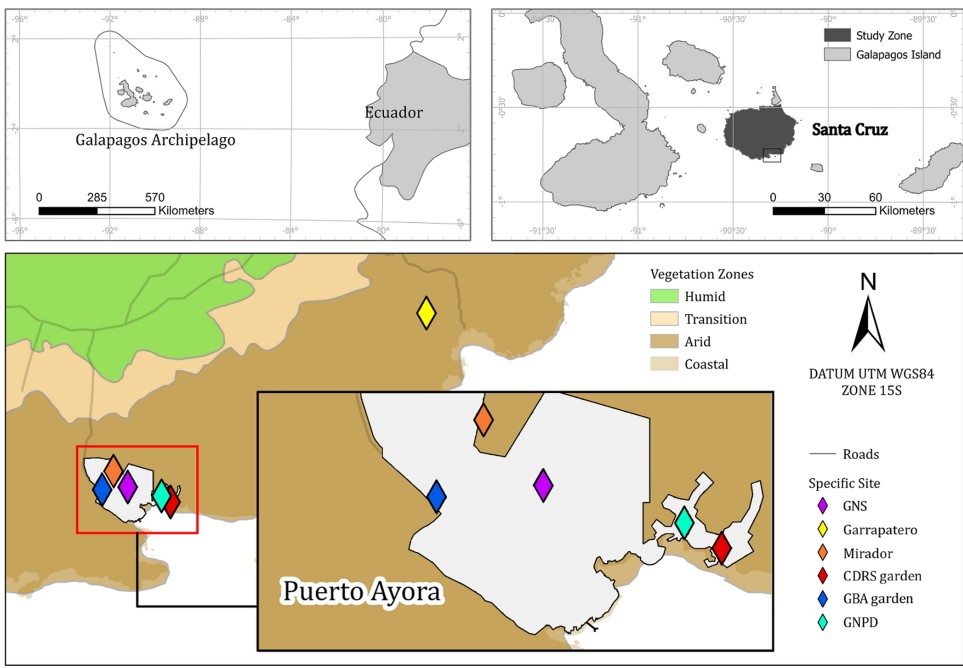

**Figure 1  Map of the *Scalesia affinis* ssp. *brachyloba* restoration study sites on Santa Cruz Island, Galapagos.** Puerto Ayora is the town in which the four garden study sites were located in. GNS, Galapagos National School; CDRS, Charles Darwin Research Station; GBA, Galapagos Biosecurity Agency; GNPD, Galapagos National Park Directorate.

either using the WST Groasis or without a WST (controls). The Groasis is a donut-shaped polypropylene tub with a lid designed to collect rainwater and store it within the tub (*Groasis®, 2019*). The bottom of the tub has a section of rope that wicks water to the area of the plant roots *via* capillary action. Additionally, the box reduces evaporation of surface water and the growth of weeds around the plant (*Jaramillo et al., 2020*). Each Groasis was

filled with 20 liters of water at the time of planting and controls were planted directly into the ground with 5 liters of water, which is the conventional maximum amount that park rangers from the Galapagos National Park use for other restoration plantings. Following planting, controls did not receive any additional water other than natural precipitation. In contrast, as per the recommended Groasis protocol, Groasis tubs were filled with 15 litres of water every three to four months from 2014 to 2018, or when the tubs had fully drained. Groasis is hypothesized to aid the survival of *S. brachyloba* by providing a constant water supply to the taproots, which mimics natural conditions of lava cracks (where the species have a potentially higher survival; *Atkinson, Jaramillo & Tapia, 2009*). This was carried out by a conservation program aimed to maximize species survival. Therefore, due to the potential benefits of using Groasis, as well as the limited resources to grow and plant seedlings of this threatened species, we decided to reduce the number of controls, leading to an unbalanced experimental design (one control used for every 6.75 individual planted with Groasis). In total between all sites, 326 plants were planted with Groasis, and 48 were planted as controls due to the higher expected mortality (*Atkinson, Jaramillo & Tapia, 2009*). Treatment was randomly assigned to each plant in the study.

Each individual plant was given a unique code and its survival monitored every three-to-four month after planting. The Groasis was removed at the point just before each plant had grown too large to fit through the center Groasis hole, approximately two years after planting. Monitoring of an individual was terminated after it died. Although the monitoring is ongoing, data presented here are from the first six years of the study: 2014–2020.

## Weather data

Weather conditions were determined from daily precipitation and temperature data available from the CDRS weather station (Charles Darwin Foundation 2020) in Puerto Ayora, Santa Cruz Island, for the period 1970–2020. These data were used to calculate total precipitation and maximum temperature across each monitoring interval (*i.e.,* approximately three to four months). Total precipitation corresponds to the sum of daily precipitation for the period assessed, and for maximum temperature, the median value was used for each interval to reduce the effect of outliers (*i.e.,* extreme values). As the time of planting was different between sites and even between plants within the site (*i.e.,* plants subsequently replaced or added), the monitoring interval was always 3-4 months, but the number of monitoring events for each plant varied.

## Effect of treatment in raw survival data

The effect of treatment (Groasis) *versus* the control was initially visualized using Kaplan–Meier (KM) survival curves (*i.e.,* predicted percentage of plants alive through time). For this, raw survival data was summarized by treatment for each planting age (*i.e.,* time since planting) with a 95% confidence interval. KM survival curves were calculated using the package 'survival' (*Therneau, 2022*) and were plotted using the package 'survminer' (*Kassambara et al., 2017*). These and all remaining statistical analyses were conducted using the R statistical language version 4.2.1 (*R Core Team, 2022*).

## Effect of treatments and other factors on seedling survival

A regression modeling approach was used to test the effect of WST treatment, total precipitation, and maximum temperature on plant survival. This was done using a repeated-measures mixed-effect logistic regression with the 'lme4' package (*Bates et al., 2015*), to model the plant survival status at each monitoring date as a function of the three factors. Survival describes whether a plant is alive or dead at the time of monitoring, while treatment indicates if the WST was used (Groasis, $n = 326$; control, $n = 48$). A full model including the interaction between treatment, total precipitation, and maximum temperature was used since the effect of the technology is expected to vary depending on temperature and precipitation. The full model was compared through a series of likelihood ratio tests (LRT) against all null models and combinations including smaller interactions and/or additive terms, through the 'anova' function in base R. All models also included planting age in days as an additive term to account for any changes in mortality due to age-related factors. To aid model convergence, maximum temperature and planting age were standardized by subtracting their mean and dividing by twice their standard deviation. Total precipitation was $\log_{10}$ transformed to scale to similar levels as the other factors. Models initially included *site* and *plant ID* as part of the random effect structure, but the sample size was too low to support this level of model complexity so *site* was dropped and only *plant ID* was included to account for the non-independent repeated measures. Although data collected from some plants covered more than 2000 days of monitoring, we analyzed data to a cap of 1352 days (3.7 years), as after that Groasis and controls had marginal drops in survival.

## Validation of model in resampled data

To test whether the effect of treatment is dependent on the unbalanced design, we used a resampling procedure where 5000 random samples without replacement of only 48 Groasis (from the 326) were compared to the controls ($n = 48$), to obtain a Chi-Square distribution statistic. For this, we re-estimate statistics of main factors through a type II Wald Chi-Squared Tests, with the function 'Anova' from the 'car' package (*Fox & Weisberg, 2018*).

## RESULTS

Plants grown with Groasis generally had a greater planting age at death than those grown without Groasis, with a median survival of at least 1.25 years (455 days, 335 to 569 95% CI) for Groasis compared to 46 days (42 to 95 95% CI) for controls (Fig. 2). This difference in survival was marked predominantly in the first two years. By 3.7 years of monitoring, both treatments had become more stable, with a three-fold higher survival of Groasis at 16% ($\pm$ 2% s.e.) compared with controls at 5.1% survival ($\pm 3.3\%$ s.e.) (Fig. 2). For controls, this corresponds to only one plant surviving, in GNP gardens (Table 1).

The selected model had a better performance than all other models tested (Table 2), and we found that all interaction effects were significant predictors of *S. brachyloba* survival [$\chi^2 = 116.28$ ($df = 7, n = 374$), $p < 0.01$ to $\chi^2 = 15.5$ ($df = 3, n = 374$), $p < 0.01$, depending on the models compared]. When controlling for all other factors we found that treatment
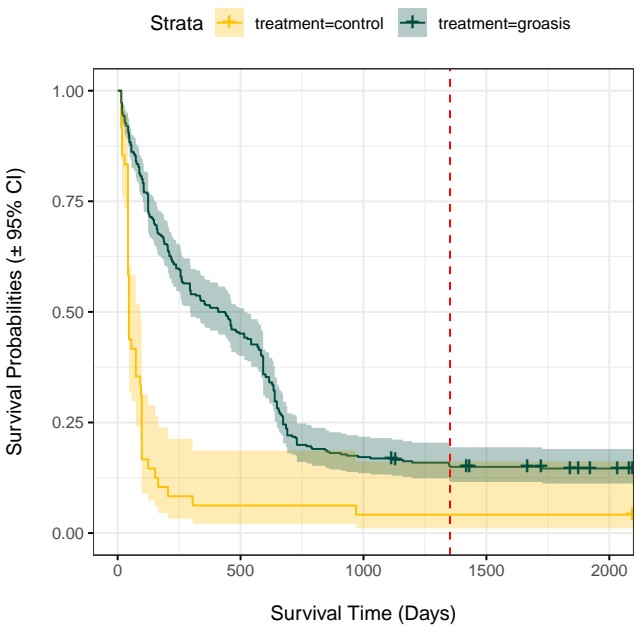

**Figure 2  Kaplan Meier survival curves for *Scalesia affinis* ssp. *brachyloba* individuals planted using Groasis technology or as controls (no technology).** Monitored period from 2014 to 2020. Red dashed line indicates the maximum planting age cap considered for analyses.

was a significant predictor of *S. brachyloba* when grown with Groasis (Fig. 3) ($\chi^2$ ($df = 1$, $n = 374$) = 48.67, $p < 0.01$), showing a high variation when grown without the technology (control). Since these effects come from a 3-way interaction model, their independent significance should be interpreted with caution, as the main effects are marginal to the full interaction. Treatment remained significant when the interaction was removed on a type II Wald Chi Square test, and when the dataset was balanced by resampling Groasis and controls (48 each) (Fig. 3; inset graph). The interaction between the three factors included in the model had an overall positive effect with odds ratio −1 of 16.585 (Table 3). When maximum temperature is segregated in two groups, total precipitation had a positive effect on predicted survival of controls (Fig. 4), particularly under conditions of 27 °C or below. On the contrary, predicted survival of plants growing with Groasis technology showed a neutral or slightly negative trend with increasing temperature and precipitation. We included planting age to account for any change due to age-related factors, and its trend was negative (Table 3, Fig. S1; coef = −0.41) but its overall effect was non-significant.

## DISCUSSION

Our results suggest that water limitation is an important factor for the early-stage survival and restoration of *S. brachyloba*. We found that water availability through precipitation increases the survival of this species and the use of Groasis consistently improved survival (Table 3, Figs. 3 and 4), probably by providing a balanced input of water. However, survival differences between control and Groasis narrowed two years after planting (Fig. 2). This

**Table 2   Likelihood ratio tests (LRT) results for the full interaction model against all model combinations.**

| Likelihood ratio test model | $\chi^2$ | df | P-value |
|---|---|---|---|
| Null Model | 116.286 | 7 | <0.001 |
| Null: treatment | 65.192 | 6 | <0.001 |
| Null: Maximum Temperature | 97.039 | 6 | <0.001 |
| Null: Total Precipitation | 121.488 | 6 | <0.001 |
| Add: Treatment + Maximum Temperature | 41.466 | 5 | <0.001 |
| Add: Treatment + Total Precipitation | 63.908 | 5 | <0.001 |
| Add: Maximum Temperature + Total Precipitation | 94.011 | 5 | <0.001 |
| Add: Full | 39.802 | 4 | <0.001 |
| Inter: Treatment × Maximum Temperature | 16.206 | 4 | 0.002 |
| Inter: Treatment × Total Precipitation | 44.006 | 4 | <0.001 |
| Inter: Maximum Temperature × Total Precipitation | 93.912 | 4 | <0.001 |
| Inter+: Maximum Temperature × Treatment + Total Precipitation | 15.500 | 3 | 0.001 |
| Inter+: Total Precipitation × Treatment + Maximum Temperature | 21.007 | 3 | <0.001 |
| Inter+: Total Precipitation × Maximum Temperature + Treatment | 39.792 | 3 | <0.001 |

Notes.

Null Model, Model without including studied factors; Null, only one factor added; Add, two factors included as additive terms; Full, all factors included; Inter, two factors included as interaction; Inter+, one interaction and one additive term.

All models include plant age as a fixed effect and plant ID as a random effect. Statistically significant values ($P < 0.05$) indicates that the more complex model (full interaction) has a better fit to simpler models.

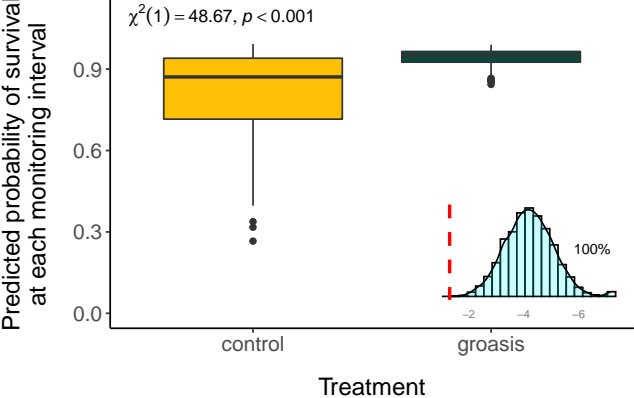

**Figure 3   Predicted survival of *Scalesia affinis* ssp. *brachyloba* individuals planted with Groasis or without it (controls) and resampled treatment significance.** Predicted survival and significance statistics between treatments are based on the full interaction model in the full dataset. In the lower-right corner, a distribution of Chi-square values for test treatment significance, based on 5,000 resamples of equal amount of Groasis as the control ($n = 48$), without considering the effect of the interaction (type II Wald Chi Square test). Dashed red line = 0.05. For visualization purpose, small $x$-axis is $\log_{10}$ transformed.

coincided with when most Groasis boxes were removed, suggesting that *S. brachyloba* plants may benefit from a longer application of this technology. Additionally, as the study was part of a large-scale restoration project framework, there were certain limitations in the experimental design because not all combinations of treatments could be applied

**Table 3** *Scalesia affinis* ssp. *brachyloba* survival as a function of treatment, total precipitation, and maximum temperature.

| Parameters | Coef. | Std. error | Z-value | Odds ratio − 1 | p-value |
|---|---|---|---|---|---|
| Treatment | 1.347 | 0.641 | 2.100 | 2.849 | 0.036 |
| TP | 0.542 | 0.494 | 1.096 | 0.720 | 0.273 |
| MT | 4.062 | 1.387 | 2.928 | 57.116 | <0.01 |
| Planting Age | −0.414 | 0.313 | −1.323 | −0.339 | 0.186 |
| Treatment × TP | −0.624 | 0.509 | −1.225 | −0.464 | 0.220 |
| Treatment × MT | −5.363 | 1.4346 | −3.738 | −0.99 | <0.01 |
| TP × MT | −2.423 | 0.949 | −2.552 | −0.911 | 0.010 |
| Full Interaction | 2.867 | 0.981 | 2.920 | 16.585 | <0.01 |

**Notes.**

Summary of logistic mixed-effect model on *Scalesia affinis* ssp. *brachyloba* survival as a function of treatment, total precipitation (TP), and Maximum Temperature (MT). MT and Plant Age were standardized, and TP was log10 transformed, prior the analyses.

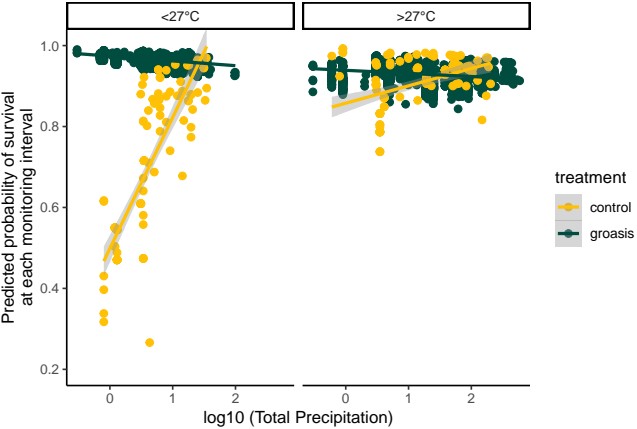

**Figure 4** **Predicted probability of survival for *Scalesia affinis* ssp. *brachyloba* as a function of total precipitation ($\log_{10}$ scaled axis) and standardized maximum temperature, since last monitoring.** For visualization purposes, data was separated according to raw temperature values (without transforming them), as "lower (or equal)" and "higher" than 27 °C.

(*e.g.*, applying the same volume of water to Groasis and controls), thus, effects of water availability presented here confounded the application of the Groasis technology.

Individuals of *S. brachyloba* usually exhibit resilience in their survival during extended periods of reduced water-availability. Having evolved with regular drought, arid-tropical species such as *S. brachyloba* may be resilient to such conditions (*Hamann, 1979b*; *Riedinger et al., 2002*). The Galapagos archipelago's location is within the El Niño Southern Oscillation (ENSO) region making it subject to drought and rainfall cycles that are especially pronounced during strong events, with some years seeing up to 3 m of annual precipitation (*Trueman & D'Ozouville, 2010*). Many plant species in Galapagos, including *Scalesia* spp., benefit from short-term increases in rainfall associated with El Niño events (*Tye & Aldáz, 1999*), as seen in other arid-tropical plant populations following an El Niño event (*Polis et*

*al., 1997*). This period is followed by a large spike in mortality of adults from long-term exposure to water *via* precipitation. This may explain why the plants grown with Groasis showed a weak negative trend with increasing precipitation (Fig. 4), indicating that the water added to the technology plus high levels of rainfall may be more than the amount that *S. brachyloba* can withstand. Nonetheless, even the lowest values of predicted survival in Groasis were close to the maximum predicted survival for controls (Fig. 4), indicating that even in extremely humid events, too little water has a more detrimental effect on *S. brachyloba* than too much water, providing a support to the value of using Groasis despite intense rainfall events.

The Galapagos hot season (*i.e.*, warmer land and sea surface temperatures from January to May and high rainfall variability) principally determines dry-zone species' annual productivity, such as that of *S. brachyloba* (*Snell & Rea, 1999*; *Trueman & D'Ozouville, 2010*; *Larrea & Di Carlo, 2011*). Our results similarly show that predicted survival was associated with rainfall, and that under lower maximum temperatures (in the June-December cold season) this interaction is more evident (Fig. 4). Although the test of temperature above or below 27 °C in this study was arbitrary and only for visualization purposes, one possible explanation is that *S. brachyloba* may be more active below certain temperature thresholds. Many tropical forest species are known to have a threshold around 27–29 °C where photosynthesis starts to decline but respiration increases (*Mau et al., 2018*). Further research addressing temperature threshold for Galapagos' plant species is needed to evaluate this hypothesis.

Overall, most high mortality rates in *S. brachyloba* appeared either after periods of cold temperature and high rainfall (>50 mm per week, maximum temperature <27 °C) (Fig. S2) or after periods of warm temperature and low rainfall (<50 mm per week, maximum temperature >27 °C), indicating two extreme conditions that may constrain species development. The link between high water availability and other *Scalesia* species' mortality has been attributed to root rot, high winds, and vine overgrowth (*Tye & Aldáz, 1999*; *Hamann, 2001*; *Larrea & Di Carlo, 2011*). While temperature and rainfall influence the species' survival, the causal mechanism was not addressed in this study. Further research is needed to establish how weather conditions directly shape *S. brachyloba* seedling survival. Research on *S. pedunculata* has shown that large-scale, rainfall-induced mortality makes way for the next cohort of young plants (*Itow & Mueller-Dombois, 1988*; *Itow, 1995*; *Tye & Aldáz, 1999*; *Hamann, 2001*; *Larrea & Di Carlo, 2011*; *Jäger et al., 2017*). Rainfall, especially during extreme weather events, may thus influence the natural regeneration cycles of related species. Similar patterns, though less pronounced, have been found in the mortality and regeneration of five dry-zone *Scalesia* species in response to the high rainfall of El Niño events (*Tye & Aldáz, 1999*; *Hamann, 2001*; *Larrea & Di Carlo, 2011*). All in all, the results of this study are informative in addressing potential planting windows for *S. brachyloba* by avoiding the most extreme conditions.

Our study only applies to the first few years of growth in young *S. brachyloba*, a period less than both the species life expectancy and ENSO events; therefore, future research should consider longer-term demographic changes in response to weather conditions. Other challenges of the current study include the large variability between sites. Evaluating

site differences were beyond the scope of the study since our goal was to extract general trends on the use of WSTs for the focal species. Nonetheless, future work should seek to further examine the importance of local conditions. For example, no plants survived more than 851 days at the Garrapatero site, which may be due to its much more rural location, where plants may be more susceptible to pest attacks (P Jaramillo, pers. obs., 2019). The overall low sample and imbalanced design is also a fundamental limitation of this study. This is largely due to the focus on conservation of a rare species that is already limited in sample size (*i.e.,* restoring *S. brachyloba*), and retrospective nature of the study (the study was conducted using data that were collected prior to a central experimental design). These limitations reduce the strength with which our results can be generalized, and it is imperative that additional prospective studies are conducted for further assessing the best tools for effectively restoring *S. brachyloba* and other species of conservation concern in Galapagos.

Galapagos arid-zone rainfall is largely determined by sea-surface temperatures, and global climate change is expected to create a wetter and warmer climate in the Galapagos (*Trueman & D'Ozouville, 2010*; *Larrea & Di Carlo, 2011*). It is more important to understand how these changes may affect endangered species in the Galapagos, especially by examining the effect of water availability and temperature on their survival and growth. This study gives insights into the effects of both factor on the early growth of *S. brachyloba* and shows promise in the use of water-saving technologies to improve survival in the face of climatic fluctuations. However, further investigations into the life cycle of *S. brachyloba* across more conditions and larger time frames are necessary to inform a more complete restoration plan for this species.

## CONCLUSIONS

The restoration of endemic species in tropical dry forest ecosystems is needed to prevent the continued degradation of these habitats (*Gillespie et al., 2013*). Water-saving technology has the potential to increase the survival of dry forest plant species, such as *Scalesia affinis* ssp. *brachyloba*. This study investigated *S. brachyloba*' relationship with two main weather conditions in arid-tropical climates: precipitation and maximum temperature, and from this, has been able to determine the effectiveness of a WST on its survival. Application of similar experimental efforts using WSTs on endangered dry forest species may produce similar positive successes, but understanding other factors that impact the species' early establishment is essential for contributing to the recovery of a globally threatened ecosystem.

## ACKNOWLEDGEMENTS

We would like to thank the Galapagos National Park Directorate, along with their park rangers both in the field and the plant nursery, especially to Danny Rueda and Rafael Chango, and the team at the GBA(Agencia de Regulación y Control de la Bioseguridad y Cuarentena para Galápagos). This paper is the result of the combined effort by all Galapagos Verde 2050 team members, especially Anna Calle-Loor, Pavel Enríquez, Paúl Mayorga,

Danyer Zambrano and David Cevallos, as well as all the local, national, and international volunteers who cared for the plants during their time with the program. We are grateful to GV2050 collaborators team and we thank the constant support of GV2050's advisors: María del Mar Trigo, James Gibbs and Washington Tapia. Finally, we would like to thank Michael Stewart, James Gibbs and Alan Tye for their helpful comments on this manuscript. The Galapagos Verde 2050 program is under DPNG investigation permission PC-21-20. This publication is contribution number 2364 of the Charles Darwin Foundation for the Galapagos Islands.

### Funding

Funding was provided by the COmON Foundation and the BESS Forest Club. There was no additional external funding received for this study. The funders had no role in study design, data collection and analysis, decision to publish, or preparation of the manuscript.

### Grant Disclosures

The following grant information was disclosed by the authors:
COmON Foundation and the BESS Forest Club.

### Competing Interests

The authors were staff or consultants of the Charles Darwin Foundation (CDF) at the time of this study. The CDF's Galapagos Verde 2050 program, of which the study was a part, was developed in consultation with the COmON Foundation (https://www.comon.earth/map-of-projects/), which provided funding for the study, is a major donor to the CDF and advises its Board. The equipment evaluated in this study was manufactured by and purchased from Groasis BV, a company independent of COmON but which supplies several COmON initiatives. The authors declare no other competing interests.

### Author Contributions

- Esme Plunkett analyzed the data, authored or reviewed drafts of the article, and approved the final draft.
- Luka Negoita analyzed the data, prepared figures and/or tables, authored or reviewed drafts of the article, and approved the final draft.
- Christian Sevilla conceived and designed the experiments, performed the experiments, authored or reviewed drafts of the article, and approved the final draft.
- Nicolás Velasco analyzed the data, prepared figures and/or tables, authored or reviewed drafts of the article, and approved the final draft.
- Patricia Jaramillo Díaz conceived and designed the experiments, performed the experiments, authored or reviewed drafts of the article, and approved the final draft.

### Field Study Permissions

The following information was supplied relating to field study approvals (*i.e.*, approving body and any reference numbers):

Galapagos National Park Directorate.

### Data Availability

The raw data is available in the Supplemental Files.

### Supplemental Information

Supplemental information for this article can be found online at http://dx.doi.org/10.7717/peerj.16367#supplemental-information.

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
