# Peer review of "Enhancing restoration success of rare plants in an arid-tropical climate through water-saving technologies: a case study of Scalesia affinis ssp. brachyloba in the Galapagos Islands"

_PeerJ, doi:10.7717/peerj.16367_

## Round 0.1 · original submission · Major Revisions

Both reviewers have raised some important issues that would need to be addressed before I can make a final decision. Also, please address the following comments:
- Please include model validation plots as supplementary material. In Fig. 4 seems like you are experiencing heteroscedasticity. This needs to be addressed: https://fukamilab.github.io/BIO202/03-C-heterogeneity.html

- I think a coefficient plot (geom_pointrange) would better present the results shown in Table 2.

- Please use colors in your map (Fig. 1).

Reviewer 1 ·

Basic reporting

Failure of restoration projects due to water limitation is high. Novel technologies are coming to market that have potential to reduce restoration costs and increase plant establishment but data is needed to support investment in them, particularly where there are small budgets. The Groasis Waterboxx technology is of particular interest for restoration of arid environments, particularly dry forest, and if effective can be used in other regions. Few papers addressing WST have delved deeper into the relationship between precipitation and survival, with and without such technologies, and these are important results to have published. This paper has potential to add important knowledge to the greater restoration literature however detail and clarify are lacking in sections. With revision it may be acceptable for publication.

The paper has professional and clear use of English language. The organization however can be improved to provide enhance clarity and strengthen the overall findings and conclusions.

Abstract
See comments on goals, results and conclusions in below and revise accordingly.

The introduction does a good job of presenting sufficient information on flora rarity on islands and the selected species and its ecological context. The first paragraph (Lines 39-47) however would benefit from pulling in more of the vast literature on 1) the challenges in arid environment (in particular dry forest) ecological restoration and 2) the use of WST in other regions. A few sentences summarizing would be sufficient, but needs to be included to identify the knowledge gaps and create the context for the paper.

Goals of paper are well stated here and are two-fold, the effect of water limitations on restoration of rare species and the potential of WST to address these limitations.

The figures included are self-explanatory, professional, albeit rather complex (see comments in next section). I was hoping to see upfront a figure or table with the mean data (with measure of variability) on survival of seedlings with and without Waterboxx technology by month or year and a figure or table on corresponding monthly precipitation during the specific monitoring period (2014-2020) so the two can be directly assessed. This should be in the paper and not supplemental.

The raw data is included but requires meta data to understand columns and rows. From methods there should be 374 individual seedlings which were monitored between 2014 and 2020. From data file, it is not possible to readily identify these 374 individuals and associated monitoring data i.e. an entry for every 3-4 months over ~6 years. Please revise.

Line specific comments
Lines 65-67. This sentence belongs in methods not introduction.
Lines 76-78. Sentence does not below in methods, but in introduction, if not already stated there.
Line 194: Define hot and cold seasons within context of study sites.
204: Define ENSO.

Experimental design

The paper meets the requirements of original research and is within the scope of the journal. The experimental design is simple and sufficient to address the research questions of interest.
Methods are not adequately described and organized so one could repeat or to fully understand the results.
• First paragraph should provide overview of ecosystem including general data on soils, vegetation, and climate. Site selection and specific information can follow.
• Separate subheadings for "Plantings and Water Technology" and "Precipitation Data" would be help with clarity in methods.
• Organize Data Analyses rather than one long paragraph. Indicate how data was manipulated and definition of measures such a survival upfront and then specific tests employed.

Specific details missing from methods include:
• Were germinated seeds planted in pots at some point in the 12 weeks, before being moved to research sites? How were they maintained during this period?
• How many controls were established at each of the 6 sites and were they randomly assigned?
• What is the rationale for only providing 5 L of water to each control seedling initially, but 20 L to the seedlings with Groasis?
• When approximately from the start of study were Groasis removed from seedlings and was it the same/similar for all seedlings and/or sites?
• Provide type and model of weather station used to collect historic and current weather data. Is location (in Puerto Ayora) representative of the Garrapatero (other side of island) and Mirador (higher elevation) sites as well?

Validity of the findings

I am not familiar with the detail of the specific indices and tests used and will assume the authors addressed all required assumptions and dealt with any issues related to the considerable unequal sample size (326 vs 48) as needed.

Tables 1 and 2 would be better as supplements and key data can be added in text. A table of mean and SE data for survival and precipitation measures I suggest would be more meaningful to the audience.

Discussion in too long and not focused. Discuss key significant findings only. Only use literature to explain a result; currently some sections seem more like a literature review and not discussion. There is also speculation and repetition of thoughts throughout. The design is simple and questions focused, so the Discussion should be as well. Revise accordingly.

Validity of finding can be better assessed in revised manuscript.

Additional comments

No additional comments.

·

Basic reporting

The project of using waterboxx to help restoration has been in place for many years and it is an important achievement to be able to monitor the effectiveness of this technique in improving survival of plantings.

The english requires some revision, there needs to be a clearer context provided. The hypothesis needs to be more clearly expressed in terms of what the data are. There needs to be more emphasis on basis data analysis and care with interpretation.

Introduction
The text needs some work. There are phrases that require clarification.
L 40 It is a non sequitur to combine species endemism and degradation. They also experience high degradation but this is not due to the high species richness
L 43 Is the dry forest flora really more threatened than the humid zone? Please clarify.
L45 A xeric plant is by definition adapted to water scarcity. I doubt that water scarcity is the key factor driving their survival. Please clarify this statement
L56-58. Here you state that the decline of Scalesia affinis has been due to habitat loss or change. This is a different threat to water scarcity.
L65. What is the goal of this study? To evaluate water availability on S affinis survival or evaluate the use of waterboxx technology on the survival of transplanted seedlings?
The introduction requires an explanation of the problems associated with restoration plantings and survivorship of newly planted seedlings. I would suggest that the potential use of this technology that needs addressing it to understand if this can reduce mortality of transplanted seedlings produced under nursery conditions and thus increase the effectiveness of restoration activities
There needs to be mention of the natural recruitment process of scalesia in the wild, and how this is linked to El Nino. This will help put the study into context.
Suggest that the WaterBoxx technology is explained in the introduction. A general comment is to standardise how you will refer to this throughout the document. You use WaterBoxx, Gorasis, WST etc

Experimental design

The research is original and fits the journal scope. The research question is meaningful but needs more careful consideration of what the expected advantage of a waterbox is, as well as whether the experimental design allows this question to be answered. Analysis requires more work, and the methods needs better clarity.

More comments
Please list in L93 how many plants in each area were used as a control. In the data set submitted I can see that the number of controls per site is the following:
ABG 1, CDF 3, Garrapatero 25, GNP 1, Mirador 18

I think that in the analysis all data from all sites were combined but I am unsure it this is justified (natural sites vs garden plantings). Is it justifiable to carry out a comparison of control and treatment when there are less than 5 control individuals in 3 of the sites. I suggest that the analysis only uses the two sites Garrapatero and Mirador which have a slightly better sample size.
Please state more clearly the watering regime. My understanding according to the text is that the controls received 5 L at planting, while the WaterBoxx received 20 litres when planted and then every 3 months received 15L for the next 6 years. However the controls did not receive any more water. Please clarify this. If this is the case, I would suggest that you need to rephrase the rationale of the study as you are comparing watering (plus waterbox) vs no water. Can we use this to determine if watering alone might help? What happens if plants without the waterbox receive water every 3 months?
If we wish to understand the waterbox effect you need to have three treatments (and ideally 4): watering without waterbox, no water or waterbox. and both. are comparing water frequency and a technology together.
Please explain in more detail planting events. The data suggest that not all plant were planted in 2014. I think this is important in data analysis. Either here or in the discussion please explain how the time of year to plant was chosen.
Results
There are several important points in the data that are not discussed due to the emphasis on the modelling (While I do not feel qualified to comment upon the modelling, I suggest that you look at some more simple aspects of the data to answer the question of whether watering/waterbox implementation helps improve seedling survival.
Here are some suggestions
1. Please show a graph which shows survivorship of waterbox and control plants over time, with different coloured bars indicating the different categories. I suggest you show Mirador and Garapatero data separately.
2. It seems from the data that not all plants were planted at the same time and that there are different planting events. If you look at survival disaggregate by planting date and site you start to see some different patterns. I suggest you present this analysis too (included below). Here you can see that there is a cohort of control plants that died after 98 days planted in 2014 and 42 days planted in 2015. This needs to be discussed
3. Why do you use a median measure? Perhaps for restoration success the most important value is mode. You want to maximize survivorship or mean.
4. The supplemental figure should be changed and added to the main text. The graphs should be divided into with and without waterboxx-additional watering. Is % Daily mortality the best measure? It appears that planting date is also important at least for the controls.
5. Is there a difference in mean survival between treatment and control and between site? At least for two sites this analysis is possible. For CDF although there is very little data it is interesting to note that the control plants have a much longer survival.

Validity of the findings

This paper shows a few important points which could be expressed more clearly
1. Waterboxx/ watering appears to increase survival of the seedlings. However it is unclear if this is the effect of the waterboxx or the regular watering episodes
2. The importance on planting date for non assisted plants. There are important differences depending on planting date in Garrapatero.
3. Low survivourship of both control and assisted plants. There appears to be a high mortality in the first 3 months in both control and treatment, and again at 1 to 2 years so that by 24 months there is a significant mortality in both categories. These two mortality events events need better explanation. I do not understand the explanation of a flooding event for the plants with waterbox technology. The data suggest that these plants died after a normal average weather pattern period, but one with a drier than average wet season.
4. Is it usual for scalesia to live only 2 years? Is this enough time for the plants to flower and set seed? If not, what is the utility of this technology? Perhaps there are alternatives?
5. Why do S affinis in a garden situation survive better for both control and treatment?
6. We know that the plant reproduction is related to el nino events. How do the mean precipitation values take into account these events?
7. This type of experiment could be carried out more easily under more controlled conditions. Has this been carried out with other species?

Additional comments

This paper needs significant revision to a) establish a clear hypothesis b) analyse the data to take into account variation in site, paucity of controls and variation in planting date and c) analyse the utility of the methods in terms of population establishment

This is a useful study to help inform post planting management. However, I suggest more emphasis is given on what the data are telling us and that the authors should not be shy of data limitations.

---

## Round 0.2 · Major Revisions

The reviewers recognize that your revised manuscript has been improved but still raise a number of major issues remaining. Please address each one of them in detail.

Reviewer 1 ·

Basic reporting

The authors have done a reasonable job with the paper revisions. The introduction and objectives and discussion are more focused, clearly written and on the most part sufficiently address my comments. The outcomes of the study, based on methods and results, need to be addressed further before accepted for publication. The paper could still use a thorough edit and I have provided some edits, but not in detail. The citation format should be reviewed as it was inconsistent throughout.

Figures are improved and now provide more succinctly the key information. Work is needed on captions to fully understand the figures on their own. See comments in document. I believe the figure captions for 3 and 4 are incorrectly paired and they should be double checked.

Experimental design

The methods provide more details and are well organized so the reader can follow what was done. However, in doing so it has brought to my attention some design issues which need to be considered when presenting results and in discussion. See below.

It would be more appropriate to use the term "planting age" instead of "plant age" throughout the paper. The plants were all 3 months (90 days) at time of planting so the plant age is not accurate.

The addition of water with Groasis Waterboxx technology compared to the control is key to the method. This has been emphasized now in a few sections and discussion and is important for interpreting results and making any recommendations. The technology allows for targeted application of water in the most critical stages of development, and once past this stage seedlings are known to have higher survival. Some literature supporting this in the introduction would be helpful.

In Study Sites section, include type of vegetation community where Scalesia affinis ssp. brachyloba is found and one is aiming to restore i.e. dry forest community. Also include mean total annual precipitation and temperature range for the region under study sites so readers not familiar with the climate have a general idea of what is average for this system.

Details of what data was recorded at each monitoring event are not provided and needs to be included in the Plantings and water-saving technology section. In the raw data there are a number of columns, including planting death age (based on monitoring date every 3-4 months) and a column for plant death age estimate. How as the death age estimated, and was this the value used in analyses?

Lines 131-136, some text to clarify why an unbalanced design was added, but it needs further clarity. I interpret that mortality without WST treatment was expected to be high, and as these were all active restoration sites, and resources to grow and plant seedlings are limited, authors chose to reduce the number of controls, thus an unbalanced design. State this more directly.

Validity of the findings

A few comments on Lines 194-199.
I do not see evidence of "stabilizing" at two years (750 days) and "leveling off" after 3.7 years (1350 days) years. These statements are contradictory too, what is the difference between stabilizing and leveling off? The controls appear to stabilize/level off around 300 days, and the treatments around 750 days (which is 2 years).
The 10x increase is in planting age at death, or end of study period for those surviving, it is not survival probability, this should be changed. If not including those still surviving, median planting age at death for treatment is 261 days and this may be more meaningful to readers. There is a 3x increase in survival between treatment and control, as noted in Abstract, and should be included here too.
Lines 197-198, rather than saying "After 3.7 of growth", reword as "After 3.7 years of monitoring" to be accurate.
As of the end of this study (June 2020) 47 treatment trees were still alive and 2 control trees. For treatment trees, half were at Mirador and all planted in 2014, and other half from the smaller sites planted between 2014 and 2017. Interestingly, none survived from Garrapatero, which was the one site located farthest from the others; none survived past 2.3 years or 851 days. If they had been analyzed separately, survival rates and median planting ages would be much higher.
These are meaningful results, not captured in the models, that could be presented in a couple sentences. In discussion can authors speculate as to why Garrapatero had such different results?

Line 232-233, does "….under lower maximum temperatures…" coincide with the period June to Dec each year? If so would be worth indicating this to understand annual variation, and assist with planting timing.

Lines 282-289, the conclusion that Groasis is a "leading tool" for restoration of the species is not substantiated by the study. The conclusion is more accurately that the technology can increase survival of the species. Survival was still very low and other factors impacting early establishment need to be determined for restoration success. This should be clear in conclusions and included in discussion. What this study's results provide in terms of understanding ideal planting timing based on precipitation and temperature is also a key outcome and needs to be included more directly in discussion and/or conclusions.

Annotated reviews are not available for download in order to protect the identity of reviewers who chose to remain anonymous.

·

Basic reporting

This version is much more focused and clearer than the previous version. However, the English still needs a lot of work. I am aware it is not the first language of many of the coauthors, but it needs serious improvement and editing (grammatical errors, erroneous use of words, and uncomfortable phrases)

Please double check the validity of the references. I picked up on one that is not in the Bibliography: Line 53-4 Myer et al 2000 is not in the reference list. Also perhaps look for other references . Gillespie et al and Tye and Fransisco Ortega are about species endemism patterns rather than threat.

Please also explain more clearly why you would expect dry forest species to be threatened by rainfall seasonality. They are adapted to this. Do you mean that rainfall seasonality is changing with climate change, or do you mean that for restoration programmes it is normal to try and increase the population yearly rather than wait for rainfall events and thus this technique could help increase seedling survival independent of rainfall?

I would suggest that the paper focusses on the use of Groasis rin the aims and leaves discussion of temperature to the and as the manipulation was based on provision of more water

Experimental design

The paper suffers from a non rigourous experimental design as the work was not set up as such, rather data was collected post hoc. I think this is legitimate in this case but the explanation on Line 134 needs better justification: Because this is a restoration project and you assumed that Groasis helped, very few plants were left without the water box, leading to an unbalanced design. It is important to mention how many controls were left in each site and use the data with care, perhaps removing data from the sites with very small N and only one control plant (as mentioned previously).

I do not feel that my comments were considered completely. Even if these comments are only addressed with additional graphs in the SI I feel it would significantly help the reader understand the analysis. Please reconsider suggestions 1,2, 4 and 5.

The text that refers to Figure 1 uses different names for the sites. Please unify this and perhaps put maps in the order of detail from left to right.

It appears that Figure 3 and 4 description and titles are reversed

Validity of the findings

Line 197 You state in the methods that Groasis was removed 2 years after planting, but in the Results you state that after 3.7 years both treatments had levelled off. I would suggest that you focus on initial differences (which is normally the most critical in restoration plantings)

---

## Round 0.3 · Minor Revisions

Please address the outstanding comments made by Reviewer 2.

Reviewer 1 ·

Basic reporting

Authors revisions have substantially improved the manuscript and I would recommend it for publication.

Experimental design

Authors revisions have substantially helped clarify methods.

Validity of the findings

Authors revisions have substantially improved the Discussion and Conclusions.

·

Basic reporting

The english is much better.

Table 1 is confusing. Please replace with a summary table that shows
Site Treatment number of plants mean (and SD) and median (and CI) age at death after transplanting,

L69 reference is quoted incorrectly. The paper states that a long tap root is a likely adaptation to surviving drought. Please change.
L83. Assisted regerenation is the normal term, not manual
L134 Suggest to say this is a conservation project aimed to maximise species survival, and measuring the utility of groasis was carried out post hoc.

Experimental design

I understand that this is a post hoc experimental design with very unbalanced design. But you do have two field sites with enough plants in the control and groasis category to analyse separately, thus controlling for site. Please include these simple statistical tests. Then perhaps you can also show there is no difference between these two sites, and I would suggest for rubustness to carry out all the analyses only with these two wild populations. It will strengthen your interpretation and you will not lose many data points.

Validity of the findings

L192. Please express median results with CI of values, not as +/- A negative number does not make sense
L195 You have one plant that survived?
L253 Suggest to look for information on perennials from arid environments, or have a look at more recent literature, as not convinced it will follow a tropical forest species

---

## Round 0.4 · accepted · Accept

Thanks for addressing the outstanding issues.